# Effect of Primary Cable Position on Accuracy in Non-Toroidal-Shaped Pass-Through Current Transformer

**DOI:** 10.3390/s24175524

**Published:** 2024-08-26

**Authors:** José M. Guerrero, Carlos A. Platero, Francisco Blázquez, José A. Sánchez

**Affiliations:** 1Electrical Engineering Department, Universidad del País Vasco/Euskal Herriko Unibertsitatea (UPV/EHU), 48007 Bilbao, Spain; josemanuel.guerrerog@ehu.eus; 2Automatic, Electric and Electronic Engineering and Industrial Computing Department, Universidad Politécnica de Madrid, 28006 Madrid, Spain; francisco.blazquez@upm.es; 3Civil, Hydraulic, Energy and Environment Engineering Department, Universidad Politécnica de Madrid, 28006 Madrid, Spain; joseangel.sanchez@upm.es

**Keywords:** cable position, current sensors, current measurement, current transformers, finite element analysis, magnetic sensors, protection transformers, sensitivity analysis, sensor testing

## Abstract

Non-toroidal-shaped primary pass-through protection current transformers (CTs) are used to measure high currents. Their design provides them with a big airgap that allow the passing of several cables per phase though them, which is the main advantage versus toroidal types, as the number of CTs required to measure the whole phase current is drastically reduced. The cables passed through the transformer window can be in several positions. As the isolines of the magnetic field generated by the primary currents are centered in the cables, if these cables are not centered in the transformer window, then the magnetic field will be non-uniform along the transformer core. Consequently, local saturations can appear if the cables are not properly disposed, causing the malfunction of the CT. In this paper, the performance of a non-toroidal-shaped protection CT is studied. This research is focused on the influence of the cable position on possible partial saturations of the CT when it is operating near to its accuracy limit. Depending on the cable position, the ratio of the primary and secondary currents can depart from the assigned ratio. The validation of this phenomenon was carried out via finite element analysis (FEA), showing that partial transformer core saturations appear in areas of the magnetic core close to the cable. By applying FEA, the admissible accuracy region for cable positioning inside the CT is also delimited. Finally, the simulation results are ratified with experimental tests performed in non-toroidal protection CTs, varying the primary cables’ positions, which are subjected to currents up to 5 kA, achieving satisfactory results. From this analysis, installation recommendations are given.

## 1. Introduction

Current transformers (CTs) are used in alternating current (AC) systems to measure extremely high currents, as the direct connection of a measurement device is not possible. One of their many applications is providing current measurements to protective relays. In order to ensure the proper operation of these relays, a dependable current measurement is needed. For this reason, there are standards that define the performance of these transformers [1,2,3].

Mainly in power system applications, when the currents are very high, several cables per phase are used. In this situation, the use of non-toroidal-shaped CTs is very convenient [4]. This is because, in high-voltage and/or high-power applications, the cables need to maintain a distance between them, and this type of CT is able, due to its geometry, to comprise all the cables of the same phase. Consequently, the number of CTs utilized per phase is reduced drastically. A theorical example of a non-toroidal CT embracing five single-phase parallel wires can be found in Figure 1. However, depending on their construction, these transformers can have performance problems. In [5], it is concluded that to avoid systematic measurement errors due to asymmetry problems in the CT, the secondary winding of the non-toroidal CT must be homogeneously distributed along the iron core.

The main cause of accuracy loss in CTs is the saturation of their magnetic core. The causes of this saturation are frequently related to the harmonic content of the primary current [6]. As current standards do not contemplate testing CTs at frequencies higher than 50 or 60 Hz, a proposal to extend their testing to higher frequencies was presented [7].

Notwithstanding this, several approaches to detecting, correcting, or compensating for magnetic core saturation effects have been proposed [8,9,10,11,12,13,14,15,16,17]. In [8], the causes of CT saturation are also linked to a large amplitude of fault currents and possible direct current (DC) components. Additionally, they implement an Improved S-Transform that allowed them to detect the saturation time and then to estimate the unsaturated AC. The primary current reconstruction from the saturation time estimation has been performed with multiple methods [9,10,11]. In [12], the inrush currents are also considered as a possible source of CT saturation. The study proposes placing CTs at both sides, primary and secondary, of a power transformer. Then, the Fréchet distance algorithm is used in both CTs, and afterwards, these are compared to each other to identify possible saturations. The comparison of CT currents at both sides of a power transformer can also be carried out with wavelet decomposition methods [13]. Zheng et al. [14] used a feature extracted from secondary current waveform histograms to distinguish whether faults were external or internal. There are other methods, such as that developed by Cavallera et al. [15], that utilize additional stray flux sensors close to the iron core to detect partial saturations in the CT if a certain threshold value is exceeded. In [16], the saturation detection is carried out by implementing an extended Kalman filter to obtain a model that allows primary unsaturated current reconstruction. Data-driven methods can also be considered to solve this issue. In [17], a deep learning approach is focused on CT saturation monitoring, based on historical and continuous monitoring data. Also, in [18], an artificial neural network (ANN) is trained to detect CT saturations during electric faults by monitoring an entire period of the secondary current waveform. Finally, the possible CT saturation due to the presence of a DC component in the primary current (because of transients or steady-state phenomena) requires a compensation method [8,16,19].

Despite the abovementioned phenomena, another source of measurement errors is the position of the primary cable relative to the magnetic core [20,21]. This effect is seldom considered in toroidal-shaped CTs [21,22], as it seems negligible compared to that commented on previously, if the cable orientation has the same direction as the normal vector of the CT radial section. Additionally, in [23], three equal Rogowski coils are tested, measuring the same cable and primary current at the same time in order to validate the measurements. From this study, it can be observed that the error between centered and displaced cables is lower than 1%. In [24], the position effect of the primary cable in toroidal yokeless Hall effect current transducers is analyzed. In this study, the error increases when the cable is separated from the transducer center up to 6.5%. This error is attenuated by increasing the number of magnetic sensors around the magnetic isoline of the transducer, i.e., increasing the radial symmetry of the transducer.

However, the cable position acquires higher importance in non-toroidal-shaped ones designed for multiple and separate cable current measurement. This is because the wire can cause partial saturations in the CT’s iron core if it is not well positioned. This effect also affects Hall effect-based magnetic sensors, in which a greater error is seen when more ellipticals are disposed among them, surrounding the primary cable [25].

Despite the position of the cable not having previously been studied in iron-core non-toroidal CTs, the main objectives of this paper are focused on the empirical analysis of the systematic error caused by this effect. Therefore, the main contribution of the study is to propose a test with which it is possible to detect partial saturations caused by the cable position in the non-toroidal CT prior to its commissioning. Thus, the correction of possible saturations during the operation of the CT is already characterized up to its accuracy limit. The analysis of this effect is performed using simulations, finite element methods (FEMs), and experimental tests carried out with non-toroidal CTs measuring up to 5 kA currents circulating through the primary cables. The results will allow us to make some recommendations to be taken into consideration in industrial applications when utilizing non-toroidal CTs. These recommendations will mainly be focused on detecting zones where primary cable positions will cause local saturations in the CTs exceeding their admissible maximum composite error.

The paper is organized as follows. In Section 2, the characteristics of the considered transformers are presented. Therefore, the simulation setup and results are presented in Section 3. Afterwards, the experimental testing arrangements and results are given in Section 4. Section 5 discusses the results obtained in the previous Sections, and finally, Section 6 summarizes the paper, highlighting the main conclusions of this research.

## 2. Materials and Methods

The performance of CTs is based on the well-known Ampère’s law; i.e., the circulation of a magnetic field along a closed loop is equal to the total current passing through the loop:(1)∮H·dl=N·i
where *H* is the magnetic field, *dl* is the length differential of the magnetic circuit, and *N*·*i* is the total current (current per wire multiplied by the number of wires). As the variables involved in this equation do not depend on material properties, it is a geometrical relationship. If there is only one cable with a circular cross section, then the system has radial symmetry. Therefore, isolines of the magnetic field are circumferences centered on the cable. In this configuration, the magnitude of this magnetic field can be easily calculated as follows:(2)$$H=\frac{N\cdot i}{2\cdot \pi \cdot r}$$
with *r* being the radius of the considered circumference.

In a CT, there are two magnetic field sources, namely, the primary current (*i*_1_) and secondary current (*i*_2_). If the secondary winding is uniformly wired around the magnetic core, then this core is a magnetic field isoline. This is due to the winding symmetry around the core cross section. So, it is an isoline for any core shape. In the case of toroidal CTs, if the primary cable is centered on the transformer window, then the magnetic nucleus is an isoline of the magnetic field created by the current carried by this primary cable. However, if the primary cable is displaced from the center, then the isolines are not concentric with the magnetic material anymore. Thus, some *H* isolines will be closed across the air gap in the farthest part of the CT. This implies that the magnetic flux density (*B*) will be more concentrated in the core part where the cable is closer. On the other hand, *B* will be lower in the core part farther from the cable, as some part of the flux lines will be concatenated through the air, becoming stray flux, *Φ_σ_*. This concept is plotted in Figure 2a. However, it is common that in conventional toroidal CTs, the air gap concatenated in the magnetic circuit will not be large enough to reach considerable flux unbalances [22]. In the case of non-toroidal current transformers, this is no longer the case, as the airgap inside the transformer is not negligible (see Figure 2b).

The performance of a CT depends on preserving the same saturation state along its measurement range. This is usually accomplished by avoiding saturation; i.e., the combined magnetic field caused by both windings has a magnetic flux density low enough to avoid saturation. This usually happens in design conditions, i.e., when the primary cable is centered on the transformer window.

Therefore, to evaluate the effect of different primary cable positions on non-toroidal CT accuracy, a protection current transformer sensor was built according to IEC 61869-2 [2]. Figure 3 shows the secondary winding and iron core of a non-toroidal CT sensor during its manufacturing process without its casing. Additionally, its characteristics are summarized in Table 1. This non-toroidal CT sensor type will be utilized as a case study in order to evaluate the effects of the primary winding position on its performance.

## 3. Simulations

The performance of the CT was simulated using Ansys Maxwell [26] FEA software (ANSYS 2020 R2). In these simulations, several primary cable positions inside the transformer core were considered.

### 3.1. Simulation Setup

With this aim, a model of the transformer core was built. Figure 4 shows the transformer core (see the cross section, the area of which is *S* = 1.43 cm^2^, on the left). The right side of this figure also shows a 2D projection of this core. The core has 154 mm long straight segments joined by semicircles with a 92 mm inner radius and 105 mm outer radius.

The transformer core is made of M-15 steel alloy [27]. Figure 5 shows its magnetizing curve. According to [2], the saturation point is calculated as the point when an increment Δ*B* = 10% requires an increment of Δ*H* = 50%. This value is reached at a flux density of *B* = 1.7 T.

To build the FEA model, not only the magnetic core but also the electric parameters should be defined. As the transformer ratio is 500/1 A and its primary has only one turn, secondary winding has around 500 turns homogeneously distributed along the magnetic core (see the turns’ disposition in Figure 3). The secondary winding is made of copper with 0.5 mm^2^ section. This implies 0.8 mm diameter. Afterwards, regarding the primary cable, the FEA simulations were performed with 240 mm^2^ cross section copper cables. This implies a 9 mm diameter.

To complete the model, a burden resistance, *R_load_* = 1 Ω, is connected to the secondary winding of the CT. This resistance represents the rated burden of the current transformer (1 VA). Therefore, the primary winding is fed through a current source. Figure 6 shows the electric scheme considered for the FEA model. This type of non-toroidal transformer has an elongated window that allows several cables in its primary circuit. For this reason, if only one cable is used, then it can be in several positions in the window. In all simulations, the current source that feeds the primary winding corresponds to the CT accuracy limit; e.g., in the CT of Table 1, it corresponds with 10 times *I*_1*N*_, 5000 A root mean square (RMS). It is modeled as a source current according to Equation (3) as follows:(3)$$i_{1}(t)=\sqrt{2}\cdot 5000\cdot sen\left(100\pi t\right)\;[A]$$

### 3.2. Simulation Results

A first set of simulations was conducted with the primary cable in different positions, which are summarized in Figure 7. Figure 8 shows the magnetic flux densities in the magnetic core for these positions. Figure 8a plots the magnetic flux density when the primary cable is in the centered position. The maximum flux density is 1.283 T, lower than the saturation flux density (see Figure 5). In this simulation, *B* has symmetry in the X-axis and the Y-axis. It can also be appreciated that both laterals of the iron core have slightly less *B* than in the upper and lower parts (Δ*B* = 0.094 T). This phenomenon is caused by the difference in reluctance for both axes. However, the symmetry in the magnetic circuit and the low B level makes the measurement error negligible.

The following Figure 8b shows the magnetic flux densities in the magnetic core when the primary cable is in the top position (65 mm above the center of the transformer window). In this case, the maximum flux density increases up to 1.642 T, close to, but lower than, the saturation flux density. Figure 8c shows the magnetic flux densities in the magnetic core when the primary cable is in the bottom position (65 mm below the center of the transformer window). The maximum *B* is 1.533 T. In both simulations Δ*B* in the Y-axis direction is considerably higher than the centered case, Δ*B* = 0.285 T, which will affect the current measurement error more. However, the maximum estimated B is still inside the saturation limits.

Finally, Figure 8d shows the magnetic flux densities in the magnetic core when the primary cable is in the right position (120 mm to the right of the center of the transformer window). The maximum flux density is 1.782 T, higher than the saturation flux density (see Figure 4) and also higher than in the vertical displacement simulations (compared with Figure 8b,c). Figure 8e shows the magnetic flux densities in the magnetic core when the primary cable is in the left position (120 mm to the left of the center of the transformer window). The maximum flux density again has the same value (*B_max_* = 1.782 T) as in its analogous simulation (see Figure 8d). In these cases, the cable has an excessive eccentricity from the ideal case.

In all cases, higher magnetic flux density levels are reached in the areas closest to the primary; meanwhile, the farthest areas have the lower flux density levels, as some of the H isolines are closed through the air and not through the iron core. This phenomenon directly affects the accuracy of the CT performance. Therefore, the higher the distance from the CT center the primary winding is, the worse the CT performance will be. Additionally, it must be remarked that the results obtained have bilateral symmetry attending to the X- and Y-axes, respectively. Consequently, analyzing only one quadrant is enough.

In order to evaluate the zone where primary cable positions provide the required accuracy, the positions shown in Figure 9 were additionally simulated. In this figure, the white area corresponds to the airgap, the gray section to the iron core, and the orange one to the casing. These consist of a mesh of cable position variations every Δ*x* = 20 mm and Δ*y* = 20 mm in the first quadrant of the air gap due to its symmetry. These increments allow most of the airgap to be covered. Additionally, they produce enough points in the mesh to study the composite error behavior in the region, and the simulations do not require extensive computation resources due to the number of points calculated not being high. The results of these simulations (secondary current RMS values) are shown in Table 2. In this table, the composite error is also given. It was calculated using Equation (4) according to [2], as follows:(4)$$\varepsilon_{c}=\frac{\sqrt{\frac{1}{T}\cdot \int_{0}^{T}{\left(R_{t}\cdot i_{2}-i_{1}\right)}^{2}dt}}{I_{1}}$$
where *Ɛ_c_* is the composite error, *R_t_* is the transformer current ratio, *I*_1_ is the primary current (in RMS), *i*_1_ and *i*_2_ are the instantaneous primary and secondary currents, respectively, and *T* is the current wave period.

From these results, it is obvious that the primary cable position has a huge effect on the current transformer accuracy reaching errors in the current measurement of more than 25% when the cables are in the farthest position from the center of the CT. The maximum admissible composite error for this CT is 5%, and it is possible to observe the limiting zone that provides the required accuracy. Figure 9 depicts this zone.

## 4. Experimental Tests

The testing procedure follows the direct testing procedure described in IEC 61869-2 [2] for protection current transformers of the P and PR class.

### 4.1. Experimental Setup

With this aim, the circuit in Figure 6 was built. Figure 10 shows the experimental test bench. The laboratory workbench uses a current source (1) capable of feeding high currents to the transformer primary (2). Several cables (2) with enough ampacity were placed in the CT (3). By feeding them sequentially, the cable position influence was studied. Two CTs (3), with parameters corresponding with those shown in Table 1, were placed near each other to corroborate that the measurement was the same in both. In the secondary current of each CT (3), a resistor (4) with *R_load_* = 1 Ω was installed. Finally, several multimeters (5) and an oscilloscope (6) were used as signal acquisitors to monitor *I*_1_, *U*_2_, and *I*_2_ in both CTs.

Afterwards, tests were conducted, controlling the transformer primary current up to its maximum current, *I*_1_ = 5 kA_RMS,_ where its accuracy limit was 5% (see Table 1). Additionally, the secondary voltage, *U*_2_, and current, *I*_2_, were measured. Several tests were performed, changing the position of the primary cable in the X-axis and in the Y-axis manually. Then, the extreme positions in both directions were evaluated. To keep the primary wires in the center, a non-ferromagnetic mount was introduced into the airgap of the CT (seen on the left side of Figure 10).

### 4.2. Experimental Results

First of all, the CT saturation performance was examined by monitoring the *I*_2_ and *U*_2_ of the CT without cables in the airgap. Thus, the saturation curve U-I was obtained by varying the *U*_2_ voltage with a controlled AC voltage source and measuring both *I*_2_ and *U*_2_. The resultant curve can be seen at Figure 11. From this figure, the saturation knee current can be obtained according to [2] at *I*_*exc*,*k*_ = 72.8 [mA] for a voltage *U*_*exc*,*k*_ = 26.1 [V]. At this point, in [2], it is stated that the composite error should be lower than the stablished limit. It can be calculated using Equation (5), as follows:(5)$$\varepsilon_{c}=\frac{I_{exc,k}}{ALF\cdot I_{2N}}$$
where *ALF* is the accuracy limit factor (*ALF* = 10 in this CT). In this case, the composite error is *ε_c_* = 0.728%, which is inside the limits (*ε_c_* < 5%) and validates the correct operation of the non-toroidal CT according to the standards. Then, a composite error of approximately *ε_c_* = 5% is expected for 10 times *I*_1*N*_.

In order to verify the influence of the position on the performance of the transformer, five different positions for the primary cable were tested. The primary cables have been fed at *I*_1_ = 5000 [A_RMS_], which implies a current of 1250 [A_RMS_] through each cable in parallel. The total number of cables across the airgap of the transformer is four, as shown in Figure 10. The positions correspond to the cable centered in the window and in the extreme bottom, top, left, and right of the CT’s air gap window. The positions were addressed by displacing the cables manually to these extremes. The aforementioned Figure 7 (in Section 3) shows these positions. These positions have been chosen in order to validate the behavior observed in the simulations, where the upper and lower cases should not experience saturation, while the right and left extremes should.

Figure 12 and Figure 13 show the secondary current, *I*_2_, waveforms compared with the primary current in terms of the secondary current, *I*_1_/*R_t_*. Both currents were measured with an oscilloscope, and their RMS values were estimated with ammeters. On the one hand, Figure 12 shows the comparison among positions in the time domain. From this figure, it can be seen that the centered position *I*_2_ measurement is the closest to the *I*_1_·*R_t_* current. The top and bottom *I*_2_ measurements are also close to the centered one. However, as previously estimated in the FEA simulations, the left and right *I*_2_ measurements do not fit well the *I*_1_/*R_t_* current. On the other hand, Figure 13 shows a comparison between *I*_2_ and *I*_1_/*R_t_* for the second quadrant positions (left and top); it can be seen that the farthest positions from the center have higher saturation values, and, as a consequence, lower accuracy levels.

From the previous measurements, the composite error was calculated using Equation (4). The RMS measured values corresponding to the positions shown in Figure 7 are collected in Table 3. The primary current *I*_1_ was set close to the accuracy limit, i.e., 5000 A_RMS_, while the *I*_2_ was collected. As expected from Figure 12 and Figure 13, the smaller *ε_c_* was given for the centered position of the cables. The top and bottom positions were still inside the accuracy limits. However, the left and right positions produced enough saturation in the iron coil to exceed the accuracy limits of the non-toroidal CT.

## 5. Discussion

The performance of our CT was evaluated through FEA simulations and experimental tests. The simulations analyzed different critical positions (*x*,*y* extreme positions inside the CT airgap and the center, see Figure 8) and a mesh of positions (see Figure 9). The tests have also been made in five extreme positions (center, top, bottom, left, and right). In the cases where there are tests and simulations, the results show a reasonable agreement, producing higher saturations in the left and right extremes. This fact validates the simulations. As a consequence, the remaining simulations allow the definition of a zone where the assigned accuracy is obtained and a zone where the primary cable should not be placed.

These results are coherent with Ampère’s law because, in a non-toroidal-shaped CT, the magnetic core is not a radial isoline for the magnetic field created by the primary current. Therefore, the further from the center the primary cable is, the greater the differences between the primary and secondary magnetic fields. So, the possibility of partial saturation in the iron core also increases. For this reason, the ratio between primary and secondary currents changes, increasing current measurement errors. Regarding the correct zone of operation, it is remarkable that this corresponds to a circle tangent to the straight parts of the inner core.

Despite partial saturation being the main phenomenon that distorts the current measurement output of the CT, its quantification depends on the iron core specifications. These specifications can vary among different CT designs. For this reason, in order to detect possible saturation phenomena in non-toroidal CTs in real facilities, a high-current injection test must be performed before commissioning. This test must be performed with the cables in the same position as they will be in the facility, and only *I*_1_ and *I*_2_ measurements are required. Thus, partial saturation will be detected by comparing both measurements, and the composite error can also be estimated for different current levels.

Finally, special attention must be given to the *i*_1_ waveform, which must be the same in the tests as in the facility once the CT is commissioned. Despite the high-current source emulating the current waveform of the primary cable of the facility, some inherent errors in the composite error estimation can appear from the facility noises, for example, due to power electronics, affecting the robustness of the method. These noises will produce additional magnetic field, which can lead to additional partial saturations. As a consequence, the estimated composite error in the tests will be lower than the error once the cable is installed in the facility. For this reason, the closer the tested *i*_1_ waveform to the facility *i*_1_, the more accurate the composite error estimation.

## 6. Conclusions

Non-toroidal-shaped primary pass-through CTs are used to measure the phase current in applications where several cables per phase, which require isolation distance, are installed. The performance of a non-toroidal-shaped current transformer for different positions of the primary cable was tested and simulated. These tests show that the accuracy is strongly dependent on the primary cable position. Ampère’s law reasoning and finite element simulations show that this result is a consequence of partial saturation in the transformer core. Therefore, the optimum performance is achieved when the primary cable is centered in the circle-delimited transformer window due to it being less prone to causing partial saturations in the CT. The further the cables from the center, the higher the composite error. Accordingly, an accuracy area was empirically defined. However, each type of non-toroidal CT must be tested individually, as this zone depends on the transformer design and primary cable current conditions.

The results of the simulations and the experimental test suggest that these current transformers should be tested using a high-current source prior to their commissioning. The high-current source must feed the CT at its maximum current, where it reaches the accuracy limit. It is strongly recommended that the same cable position be used in the tests and in the electric plant where it will be installed. In other words, the current distribution during the tests should be the same as in the operation of the current transformer. In this way, the operators could monitor possible local saturations during the tests and quantify the composite error before CT commissioning, which must never exceed its accuracy limit. Additionally, special attention should be paid to the short-circuit current, as the saturation problem will be achieved in these operation conditions.

The authors are working, as a future work, on the grounding connection of the shields of the high-voltage cable in this type of current transformer.

## Figures and Tables

**Figure 1 sensors-24-05524-f001:**
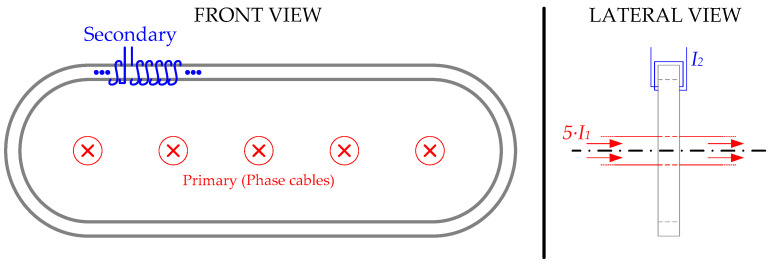
Non-toroidal current transformer application—graphical example.

**Figure 2 sensors-24-05524-f002:**
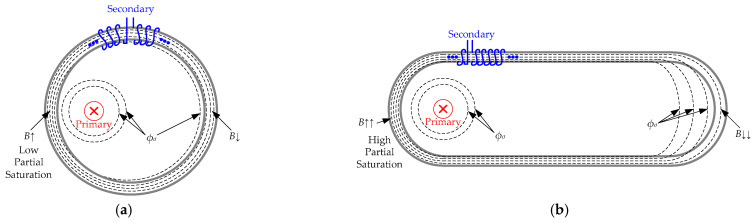
Current transformer theoretical flux line distribution for a non-centered primary cable: (**a**) Toroidal shape, lower local saturation, (**b**) non-toroidal shape, higher local saturation.

**Figure 3 sensors-24-05524-f003:**
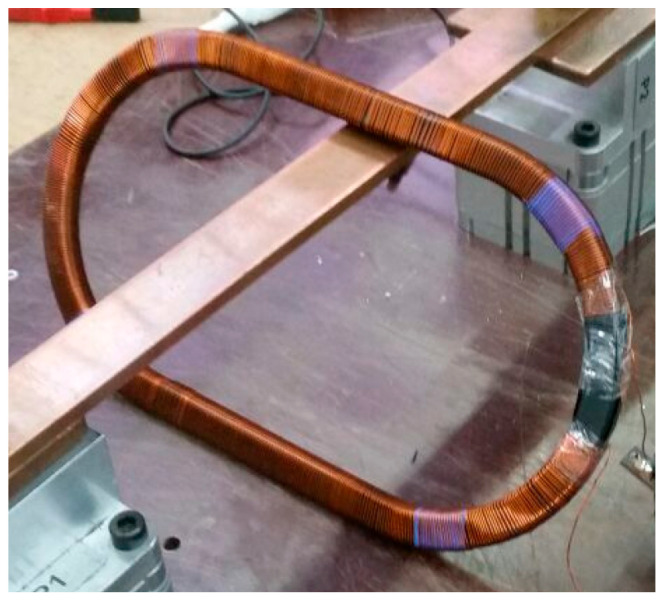
Case study: manufactured non-toroidal CT iron core and secondary winding.

**Figure 4 sensors-24-05524-f004:**
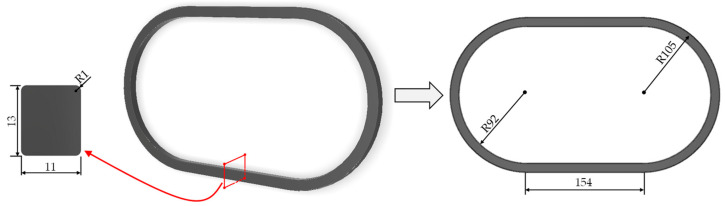
Magnetic core layout and dimensions.

**Figure 5 sensors-24-05524-f005:**
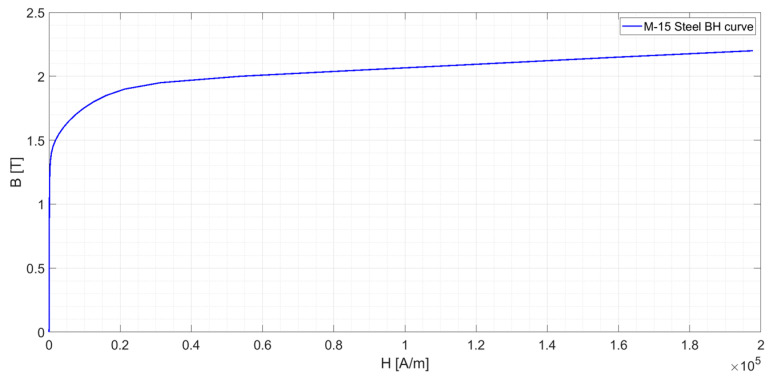
Magnetization curve of M-15 steel.

**Figure 6 sensors-24-05524-f006:**
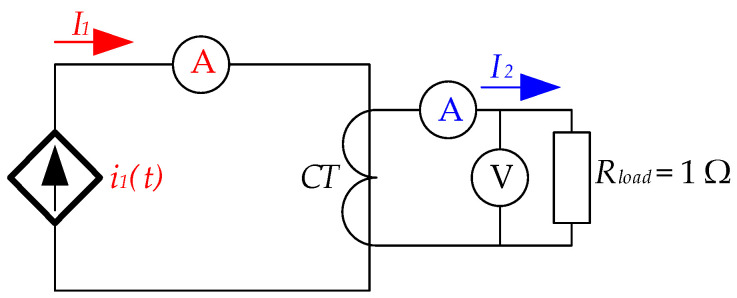
Electrical scheme considered in simulations.

**Figure 7 sensors-24-05524-f007:**
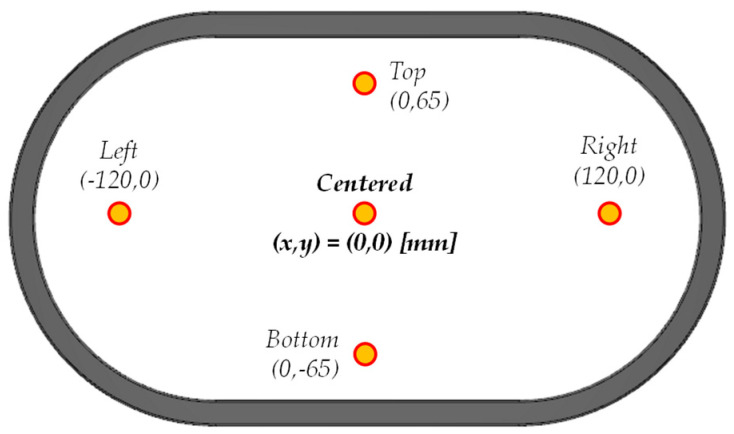
Simulated cable positions inside the non-toroidal CT.

**Figure 8 sensors-24-05524-f008:**
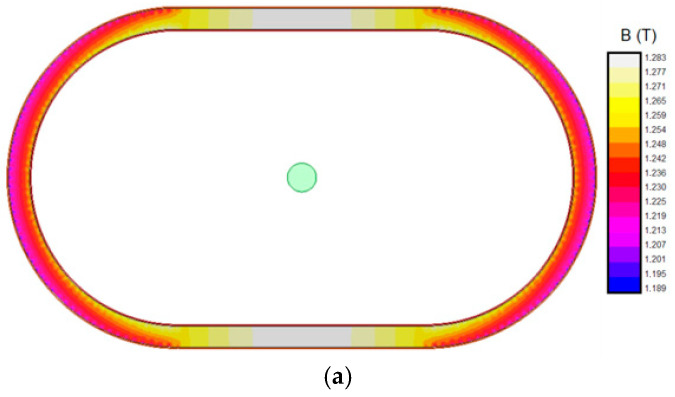
Simulations of the magnetic flux density in the CT transformer. (**a**) Primary cable centered in the transformer window (0 mm, 0 mm); (**b**) primary cable on the upper part of the transformer window (0 mm, 65 mm); (**c**) primary cable on the lower part of the transformer window (0 mm, 65 mm); (**d**) primary cable on the right part of the transformer window (120 mm, 0 mm); (**e**) primary cable on the left part of the transformer window (−120 mm, 0 mm).

**Figure 9 sensors-24-05524-f009:**
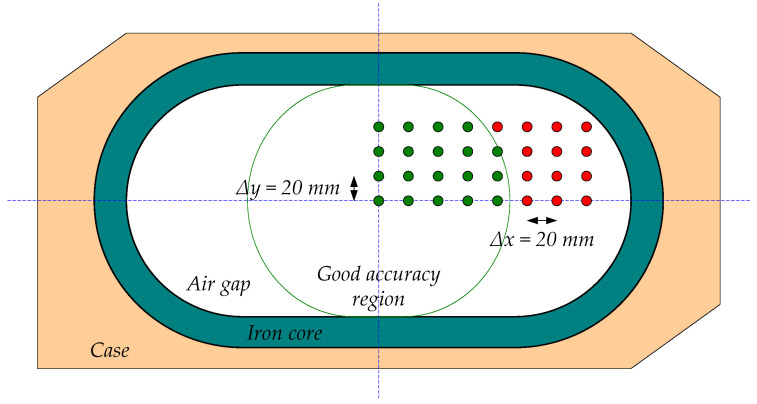
Simulated mesh of primary cable positions with Δx = Δy = 20 mm. The primary cable position zone for admissible accuracy performance in the non-toroidal CT (green points correspond to a low enough ratio error, and red points correspond to an excessive ratio error; orange area, casing; gray area, iron core; white area, airgap; green area, admissible error zone of primary winding positioning).

**Figure 10 sensors-24-05524-f010:**
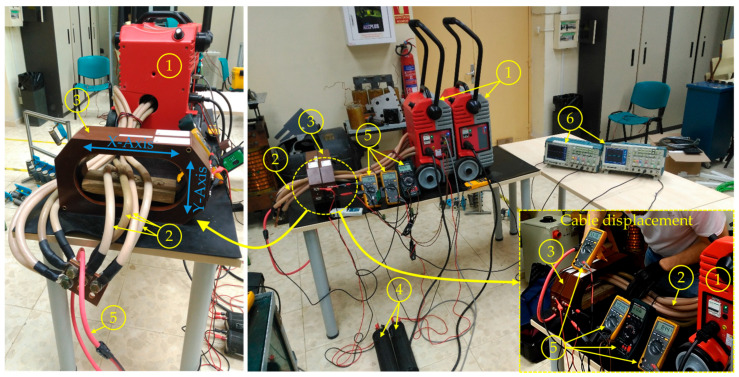
Laboratory test setup—front and lateral views: (1) current source; (2) primary high power cables; (3) tested non-toroidal CTs; (4) secondary resistive loads; (5) *U*_2_, *I*_2_, and *I*_1_ measurements; (6) oscilloscopes.

**Figure 11 sensors-24-05524-f011:**
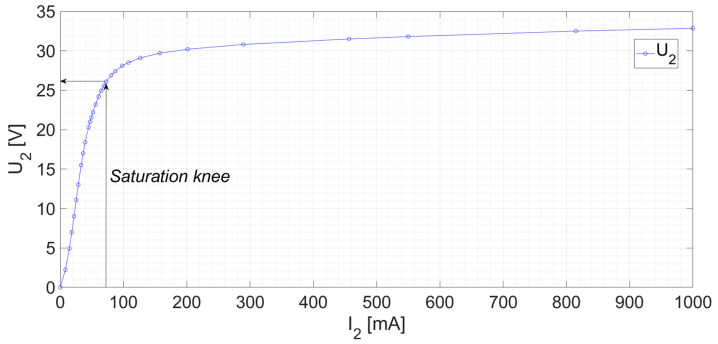
CT saturation curve. Voltage *U*_2_ and Current *I*_2_ of the secondary winding with the primary in open circuit.

**Figure 12 sensors-24-05524-f012:**
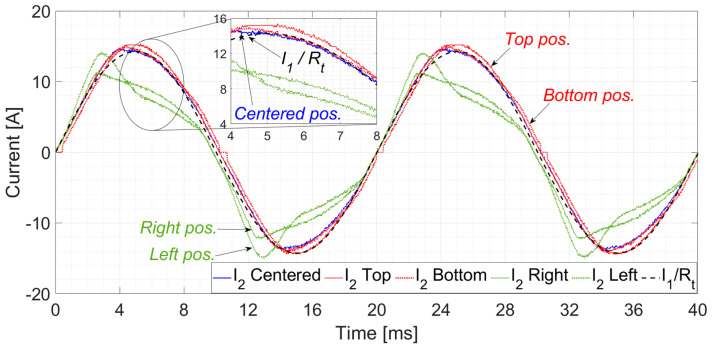
Secondary wave currents, *i*_2_, corresponding to the five tested positions inside the air gap of the CT and the primary current wave converted to the secondary current of the CT, *R_t_*·*i*_1_.

**Figure 13 sensors-24-05524-f013:**
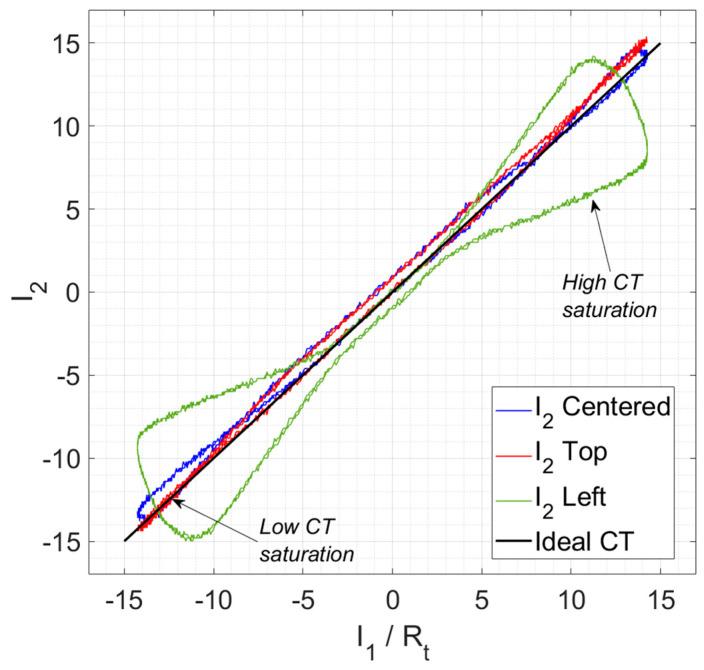
Observation of the CT saturation for the centered, top, and left primary cable positions comparing *i*_1_/*R_t_* and *i*_2_.

**Table 1 sensors-24-05524-t001:** Current transformer sensor characteristics.

Parameter		Magnitude	Unit
Manufacturer		ELEQ	
Model		GSL235 × 420 A 155 × 310	
Rated primary current	*I* _1*N*_	500	A
Rated secondary current	*I* _2*N*_	1	A
Rated burden power	*S_N_*	1	VA
Accuracy class		5P10	
Frequency	*f*	50	Hz
Rated voltage	*U_N_*	0.72/3/-	kV
Insulation thermal class		E	
Short-time thermal current		50/1	kA/s
Operating temperature range	*T*	−5/+40	°C

**Table 2 sensors-24-05524-t002:** CT-FEA simulation results of *I_2_* and *Ɛ_c_* for different meshed positions.

		Secondary Current, *I*_2_ [A_RMS_]/*I*_1_ = 5000 [A_RMS_]							
		X Distance to the Center (mm)							
		0	20	40	60	80	100	120	140
Y Distance (mm)	0	9.91	9.88	9.87	9.87	9.84	9.74	9.45	8.77
	20	9.88	9.88	9.87	9.87	9.83	9.72	9.41	8.70
	40	9.87	9.87	9.87	9.86	9.80	9.65	9.28	8.47
	60	9.87	9.87	9.86	9.82	9.74	9.52	9.02	8.02
		**Composite Error, *Ɛ_c_* (%)**							
		**Horizontal Distance to the Center (mm)**							
		**0**	**20**	**40**	**60**	**80**	**100**	**120**	**140**
Y Distance (mm)	0	2.33	3.21	3.23	3.36	4.10	6.73	14.01	31.08
	20	3.21	3.21	3.24	3.42	4.29	7.18	14.96	32.87
	40	3.23	3.24	3.32	3.68	4.98	8.76	18.14	38.69
	60	3.36	3.42	3.68	4.49	6.73	12.27	24.66	50.03

**Table 3 sensors-24-05524-t003:** Current transformer test results for different primary cable positions.

Primary Cable Position	*I*_1_ [A_RMS_]	*I*_2_ [A_RMS_]	Composite Error (ε_c_)
Center	5043	9.96	3.81%
Left	5024	8.17	47.61%
Right	5045	8.17	47.62%
Top	5027	9.89	4.39%
Bottom	5069	9.93	4.38%

## Data Availability

Data are available under request.

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
