# Peer review of "Effect of Primary Cable Position on Accuracy in Non-Toroidal-Shaped Pass-Through Current Transformer"

_sensors, 2024, doi:10.3390/s24175524_

Round 1
Reviewer 1 Report
Comments and Suggestions for Authors
The study investigates the effects of magnetic saturation on the measurement accuracy of current transformers (CTs) in high-voltage power cables. It focuses on how different positions of the main cable and varying current levels impact the CTs' performance and measurement errors. The study is well-organized, and the results are relatively clear. Nevertheless, the manuscript still contains numerous areas that require clarification and improvement.
1. The study indicates that magnetic saturation at different positions of the main cable can cause measurement errors. However, it lacks detailed information on the specific saturation levels at these positions. To fully assess the impact on measurement errors, please provide detailed information on the magnetic saturation levels at specific positions of the main cable and further explain the impact of each level on measurement errors.
2. The description of the simulation and experimental methods in the study does not clarify the criteria and rationale for selecting specific positions of the main cable for testing. Why these positions were chosen and whether they adequately represent to ensure the objectivity and accuracy of the results? Please explain the rationale.
3. The study mentions that current transformers must be tested with high current sources but does not specify the exact current levels used in the experiments. Identifying the specific current levels is crucial to accurately replicate real-world operating conditions and to help readers better understand the testing conditions.
4. The conclusion of the study indicates that optimal performance is achieved when the main cable is positioned centrally. However, the study does not provide detailed information on the specific size of this optimal operating region. Therefore, detailed information on the size and limits of the optimal operating region is needed.
Comments on the Quality of English LanguageThe text generally maintains good grammatical structure, but a few sentences could be clearer with minor revisions.
Author Response
For research article:
“Primary Cable Position Effect on Non-Toroidal Shape Pass Through Current Transformer Accuracy”
Paper ID: Sensors-3144466
|
Response to Reviewer 1 Comments
|
||||||||||||||||||||||||||||||||||||||||||||||||||||||||||||||||||||||||||||||||||||||||||||||||||||||||||
|
1. Summary |
|
|
||||||||||||||||||||||||||||||||||||||||||||||||||||||||||||||||||||||||||||||||||||||||||||||||||||||||
|
First of all, we would like to thank you for your time and effort spent in the revision process of this manuscript. We have carefully read, analyzed and considered all the comments and suggestions you have provided. Please, find the detailed responses below in this document attached as well as the manuscript changes, which have been yellow highlighted. With these modifications, we think the paper is clearer and its quality has considerably improved.
|
||||||||||||||||||||||||||||||||||||||||||||||||||||||||||||||||||||||||||||||||||||||||||||||||||||||||||
|
2. Questions for General Evaluation |
Reviewer’s Evaluation |
Response and Revisions |
||||||||||||||||||||||||||||||||||||||||||||||||||||||||||||||||||||||||||||||||||||||||||||||||||||||||
|
Does the introduction provide sufficient background and include all relevant references? |
Must be improved |
Thank you for these valuable and positive evaluation about the manuscript. We have used it to improve the revised version of the manuscript. Please, find the corresponding answers to these comments in the “Point-by-point” section of this reply, and if any of the points remain unclear or non-fulfilled, please do not hesitate to tell us. |
||||||||||||||||||||||||||||||||||||||||||||||||||||||||||||||||||||||||||||||||||||||||||||||||||||||||
|
Is the research design appropriate? |
Can be improved |
|||||||||||||||||||||||||||||||||||||||||||||||||||||||||||||||||||||||||||||||||||||||||||||||||||||||||
|
Are the methods adequately described? |
Must be improved |
|||||||||||||||||||||||||||||||||||||||||||||||||||||||||||||||||||||||||||||||||||||||||||||||||||||||||
|
Are the results clearly presented? |
Can be improved |
|||||||||||||||||||||||||||||||||||||||||||||||||||||||||||||||||||||||||||||||||||||||||||||||||||||||||
|
Are the conclusions supported by the results? |
Must be improved |
|||||||||||||||||||||||||||||||||||||||||||||||||||||||||||||||||||||||||||||||||||||||||||||||||||||||||
|
|
|
|
||||||||||||||||||||||||||||||||||||||||||||||||||||||||||||||||||||||||||||||||||||||||||||||||||||||||
|
|
|
|
||||||||||||||||||||||||||||||||||||||||||||||||||||||||||||||||||||||||||||||||||||||||||||||||||||||||
|
3. Point-by-point response to Comments and Suggestions for Authors |
||||||||||||||||||||||||||||||||||||||||||||||||||||||||||||||||||||||||||||||||||||||||||||||||||||||||||
|
Comments 1: The study investigates the effects of magnetic saturation on the measurement accuracy of current transformers (CTs) in high-voltage power cables. It focuses on how different positions of the main cable and varying current levels impact the CTs' performance and measurement errors. The study is well-organized, and the results are relatively clear. Nevertheless, the manuscript still contains numerous areas that require clarification and improvement.
The study indicates that magnetic saturation at different positions of the main cable can cause measurement errors. However, it lacks detailed information on the specific saturation levels at these positions. To fully assess the impact on measurement errors, please provide detailed information on the magnetic saturation levels at specific positions of the main cable and further explain the impact of each level on measurement errors.
|
||||||||||||||||||||||||||||||||||||||||||||||||||||||||||||||||||||||||||||||||||||||||||||||||||||||||||
|
Response 1: Thank you for this comment. It is an interesting topic.
On the one hand, we have pointed out that the magnetic saturation of the iron core for different positions of the CT primary cable is one of the errors that these measurement devices can experience. However, the magnetic saturation description has been done in a generic way, as each CT strongly depends on the magnetic lamination qualities of their iron-core. Consequently, the saturation levels of B are different in each transformer.
The results of the present study are only valid for the tested series of CTs. As its saturation level is set at 1.7 T. For different current transformers specifications, the area with acceptable composite error could be different, but the conclusions achieved in this manuscript can be applied to any non-toroidal shape CT.
This is the reason why we strongly recommend performing the tests before commissioning the CT with the cables in the same position as they will be installed in the facility in order to detect possible partial saturations of the CT.
On the other hand, the partial saturation level of a CT is a very difficult task to measure in real facilities. But with the procedure with are considering only current measurements are required in both sides of the CT (i1 and i2) to observe if the composite error is above or below the limits. These magnitudes are very easy to measure with ammeters in industry and complex procedures are avoided.
Considering both ideas, we had not gone into the partial saturation in depth in order to make it easier to understand for readers. Moreover, the partial saturation effect can be only observed in Fig. 8 for the left and right positions (outside the acceptable composite region of Fig. 9). The centered, upper and lower positions do not present partial saturation at all. Afterwards, this partial saturation can be observed indirectly in the current measurement of the experimental tests (see Fig. 12 and Fig. 13 of the paper).
In order to allude this issue in the paper we have included the following paragraph at the end of Section 5: Discussion commenting that the partial saturation depends on the CTs design and that they can be easily detected by measuring both the primary and secondary currents. It states as follows: “Despite the partial saturation is the main phenomenon that distorts the current measurement output of the CT, its quantification depends on the iron specifications. These specifications can vary among different CT designs. For this reason, in order to detect possible saturation phenomena in non-toroidal CTs of real facilities a high current injection test must be performed before its commissioning. The test must be performed with the cables in the same position as they will be in the facility and only I1 and I2 measurements are required. Thus, partial saturation will be detected comparing both measurements, and also, the composite error can be estimated for different current levels.”
We hope the issue will be clearer now. Thank you for pointing out it. |
||||||||||||||||||||||||||||||||||||||||||||||||||||||||||||||||||||||||||||||||||||||||||||||||||||||||||
|
Comments 2: The description of the simulation and experimental methods in the study does not clarify the criteria and rationale for selecting specific positions of the main cable for testing. Why these positions were chosen and whether they adequately represent to ensure the objectivity and accuracy of the results? Please explain the rationale.
|
||||||||||||||||||||||||||||||||||||||||||||||||||||||||||||||||||||||||||||||||||||||||||||||||||||||||||
|
Response 2: The criteria in experimental tests and simulations for selecting specific positions of the primary cable was to perform a measurement every 20 mm in both X-Y directions. This way, a mesh of positions is built, allowing us to evaluate the position effect in the non-toroidal current transformer.
We humbly think that 20 mm in every direction should be enough to define an operation zone with a good resolution and low computational resources.
Additionally, the mesh was only built in the first quadrant of the iron core, as it was previously demonstrated in Fig. 8 that the CT behavior had symmetry in both X-Y axes.
On the other hand, in the experimental tests we only did the extreme positions corresponding to the left, right, upper and lower positions and the centered one to validate the simulation results.
However, as you pointed out, we have carefully reread the manuscript and we have realized that this issue should be clarified in the paper. In order to address it, we have introduced the following modifications in the manuscript (yellow highlighted):
Section 3.2: Simulation Results: “In order to evaluate the zone where primary cable positions provide its required accuracy, the positions shown in Figure 9 were additionally simulated. In this figure, the white area corresponds to the airgap, the grey section to the iron core and the orange one to the casing. They consist of a mesh of cable position variations every Δx = 20 mm and Δy = 20 mm in the first quadrant of the air gap due to its symmetry. These increments allow covering most part of the airgap. Additionally, they produce enough points in the mesh to study the composite error behavior in the region, and the simulations do not require extensive computation resources due to not a big number of points are calculated. The results of these simulations (secondary current RMS values) are shown …”
Section 4.2: Experimental Results: “In order to verify the influence of the position in the performance of the transformer five different positions of the primary cable were tested. The primary cables have been fed at I1 = 5000 [ARMS] , that implies a current of 1250 [ARMS] through each cable in parallel. The total number of cables across the airgap of the transformer is 4 as it was shown in Figure 10. The positions correspond to the cable centered in the window, and in the extremes low, high, left and right of the CT’s air gap window. The positions were addressed by displacing the cables manually to these extremes. The aforementioned Figure 7 (see in Section 3) shows these positions. These positions have been chosen in order to validate the behavior observed in the simulations, where the upper and lower cases should not experience saturation, while the right and left extremes should.”
Additionally, we have modified the conclusions section to also highlight that, despite our tested positions, the recommended tested positions are the ones they are going to be used in the real facility once the CT is commissioned. Then, we have modified Section 6: Conclusions as follows: “The results of the simulations and the experimental test suggest that these current transformers should be tested using a high current source previously their commissioning. The high current source must feed the CT at its maximum current where it reaches the accuracy limit. It is strongly recommended that the same cables position should be used in the tests and in the electric plant where it will be installed. In order words, the current distribution during the tests should be the same as in the operation of the current transformer. In such manner, the operators could monitor possible local saturations during the tests and quantify the composite error before the CT commissioning, that must never exceed its accuracy limit. Additionally, especial attention should be paid to the short-circuit current, as the saturation problem will be achieved in this operation conditions.”
We hope that, with these modifications, the rationale will be clearer. Thank you very much again. |
||||||||||||||||||||||||||||||||||||||||||||||||||||||||||||||||||||||||||||||||||||||||||||||||||||||||||
|
Comments 3: The study mentions that current transformers must be tested with high current sources but does not specify the exact current levels used in the experiments. Identifying the specific current levels is crucial to accurately replicate real-world operating conditions and to help readers better understand the testing conditions.
Response 3: Dear reviewer, thank you very much for this comment as it is very important issue. The tests should be performed at the maximum current. For example, in our transformer the accuracy class is 5P10, that means that the current transformer has a 5 % error at 10 times the rated current. The current ratio is 500/1 A, therefore the current transformer should be tested at 5000 A (10x500A). We already said it in the manuscript in Section 3.1: Simulation Setup as follows:
“… In all simulations, the current source that feeds the primary winding corresponds to the accuracy limit, 5000 A Root Mean Square (RMS). It is modeled as a source current according to (3) as:
and in Section 4.1: Experimental setup as:
“Afterwards, tests have been made controlling the transformer primary current up to I1 = 5 kARMS. …”
In order to clarify this issue, we have modified both sentences respectively as follows: In Section 3.1: “ … In all simulations, the current source that feeds the primary winding corresponds to the CT accuracy limit, e.g., in the CT of Table 1, it corresponds with 10 times I1N, 5000 A Root Mean Square (RMS). It is modeled as a source current according to (3) as: …”
And in Section 4.1: “Afterwards, tests have been made controlling the transformer primary current up to its maximum current, I1 = 5 kARMS, where its accuracy limit is 5% (see Table 1). …”
Additionally, we have clarified it in the Section 6: Conclusions as follows: “The results of the simulations and the experimental test suggest that these current transformers should be tested using a high current source previously their commissioning. The high current source must feed the CT at the maximum current where it reaches the accuracy limit. It is strongly recommended that …”
We apologize for the possible misunderstanding and we expect that the issue will be clearer now with the modifications given. But if the problem remains, please do not hesitate to tell us. Thank you. |
||||||||||||||||||||||||||||||||||||||||||||||||||||||||||||||||||||||||||||||||||||||||||||||||||||||||||
|
Comments 4: The conclusion of the study indicates that optimal performance is achieved when the main cable is positioned centrally. However, the study does not provide detailed information on the specific size of this optimal operating region. Therefore, detailed information on the size and limits of the optimal operating region is needed.
|
||||||||||||||||||||||||||||||||||||||||||||||||||||||||||||||||||||||||||||||||||||||||||||||||||||||||||
|
Response 4: As you pointed out, we concluded in the paper that the optimal performance is achieved when the cable is in the center of the airgap. This conclusion as well as the operation zone were determined empirically, as it can be observed from the results collected in Table 2 (Simulation results):
and Table 3 (experimental results):
where for the same primary currents, the composite error is lower. It is a general recommendation due to the centered position is less prone to cause partial saturations in the CT than other positions. Additionally, it has been observed from simulations that the area corresponds to a circle due to the X and Y axes have symmetry, respectively. However, this type of current transformers should be studied individually, because their accuracy zone will depend on the CT design and primary cable current conditions.
It is important to remark that after our experience, the most important task is to test the current transformer injecting the primary current at the same position as it will operate in the switchgear/industrial facility. This way, the accuracy of the CT could be validated and possible design problems could be diagnosed due to local saturations at high current levels previously the transformer commissioning.
In order to clarify this issue, we have modified both paragraphs of Section 6: Conclusions as follows:
First paragraph modifications are focused on the accuracy area definition issue: “Non-toroidal shape primary pass through CTs are used to measure the phase current in applications where several cables per phase, that need isolation distance among them, are installed. The performance of a non-toroidal shape current transformer for different positions of the primary cable have been tested and simulated. These tests show that accuracy is strongly dependent on primary cable position. Ampere’s law reasoning and finite element simulations made show that this result is consequence of partial saturation in the transformer core. Therefore, optimum performance is achieved when primary cable is centered in circle-delimited transformer window due to it is less prone to cause partial saturations in the CT. The farther the cables are from the center, the higher the composite error is. Attending to this, an accuracy area was empirically defined. However, each type of non-toroidal CT must be tested individually, as this zone depends on the transformer design and primary cable current conditions.”
Meanwhile, second paragraph modifications are focused on the cable position testing problem: “The results of the simulations and the experimental test suggest that these current transformers should be tested using a high current source previously their commissioning. The high current source must feed the CT at its maximum current where it reaches the accuracy limit. It is strongly recommended that the same cables position should be used in the tests and in the electric plant where it will be installed. In order words, the current distribution during the tests should be the same as in the operation of the current transformer. In such manner, the operators could monitor possible local saturations during the tests and quantify the composite error before the CT commissioning, that must never exceed its accuracy limit. Additionally, especial attention should be paid to the short-circuit current, as the saturation problem will be achieved in this operation conditions.”
|
||||||||||||||||||||||||||||||||||||||||||||||||||||||||||||||||||||||||||||||||||||||||||||||||||||||||||
|
4. Response to Comments on the Quality of English Language |
||||||||||||||||||||||||||||||||||||||||||||||||||||||||||||||||||||||||||||||||||||||||||||||||||||||||||
|
Point 1: The text generally maintains good grammatical structure, but a few sentences could be clearer with minor revisions.
|
||||||||||||||||||||||||||||||||||||||||||||||||||||||||||||||||||||||||||||||||||||||||||||||||||||||||||
|
Response 1: Thank you for this comment. We have performed an extensive English review in order to check and fix the sentences that you pointed out. You will find the changes yellow highlighted along the text. |
||||||||||||||||||||||||||||||||||||||||||||||||||||||||||||||||||||||||||||||||||||||||||||||||||||||||||
|
|
||||||||||||||||||||||||||||||||||||||||||||||||||||||||||||||||||||||||||||||||||||||||||||||||||||||||||
We would like to thank you again for your positive and constructive comments and suggestions. We hope all your concerns will be properly addressed.

Reviewer 2 Report
Comments and Suggestions for Authors
The local saturation of non-toroidal shape current transformer is one of the key factors to influence its accuracy, especailly under the condition of high currents or short currents. The phenomenon has been studied with simulation and experiment. The conclusions could provide valuable references for the engineering applications of not-toroidal shape currents transformers. However, the study has obvious shortage in the innovation and creativity.
Reviewer 3 Report
Comments and Suggestions for Authors
1.Please introduce the recent research progress in this field, and, compared with the latest research content, please elaborate on the advantages of this method .
2.The paper needs language correction. such as the sentence in line 221 lacks the verb of "are" after “i2”.
3.In some aspects, such as the application domain, whether this method has limitations?
4.What are the authors' future work in their research focusing on this method?
5.Please discuss the robustness of the proposed method.
Comments on the Quality of English Language
The paper needs some language correction.
Round 2
Reviewer 1 Report
Comments and Suggestions for Authors
The author has addressed all issues. The paper is now suitable for publication.
Reviewer 2 Report
Comments and Suggestions for Authors
The magnetic saturation of current transformer is one of the key factors to influence its accuracy. The research is focused on the cable position influence in possible partial saturations of the non-toroidal shape CT when it is operating near to its accuracy limit. The performance of a non-toroidal shape protection CT is studied with simulation and experiment. The main contribution is clearly expressed in the modified version. The conclusions could provide valuable references for the engineering applications and theoretical study of not-toroidal shape currents transformers.